# Toxicity Analysis of Mesoporous Polydopamine on Intestinal Tissue and Microflora

**DOI:** 10.3390/molecules27196461

**Published:** 2022-09-30

**Authors:** Luoyijun Xie, Qiyan Li, Yingying Liao, Zihua Huang, Yulin Liu, Chutong Liu, Leilei Shi, Qingjiao Li, Miaomiao Yuan

**Affiliations:** The Eighth Affiliated Hospital, Sun Yat-sen University, Shenzhen 518017, China

**Keywords:** intestinal microflora, mesoporous polydopamine, toxicity analysis, 16S rRNA gene sequencing

## Abstract

As a promising therapy, photothermal therapy (PTT) converts near-infrared (NIR) light into heat through efficient photothermal agents (PTAs), causing a rapid increase in local temperature. Considering the importance of PTAs in the clinical application of PTT, the safety of PTAs should be carefully evaluated before their widespread use. As a promising PTA, mesoporous polydopamine (MPDA) was studied for its clinical applications for tumor photothermal therapy and drug delivery. Given the important role that intestinal microflora plays in health, the impacts of MPDA on the intestine and on intestinal microflora were systematically evaluated in this study. Through biological and animal experiments, it was found that MPDA exhibited excellent biocompatibility, in vitro and in vivo. Moreover, 16S rRNA analysis demonstrated that there was no obvious difference in the composition and classification of intestinal microflora between different drug delivery groups and the control group. The results provided new evidence that MPDA was safe to use in large doses via different drug delivery means, and this lays the foundation for further clinical applications.

## 1. Introduction

In recent decades, photothermal therapy (PTT) has demonstrated great potential as a cancer treatment approach [1]. As the essential element in PTT, photothermal agents (PTAs) absorb energy from lasers and release it partially in the form of heat to destroy tumor cells. Given the potential of PTT for clinical applications, the safety of PTAs need to be thoroughly verified before their widespread use in clinical treatments.

As a common PTA in clinical research, mesoporous polydopamine (MPDA) has superior near-infrared (NIR) responsive properties and a high photothermal conversion efficiency [2]. Zhang et al. [3] reported that a PEG-modified MPDA carrier loaded with paclitaxel (PTX) effectively inhibited tumor growth and it also exhibited better synergy therapeutic effects. Moreover, photothermal therapy caused by MPDA also induced tumor cells to expose many tumor antigens to promote dendritic cell (DC) maturation, and activate the T cell immune response, according to Xiao Zheng’s report [4]. Additionally, MPDA also was found to be an organic polymer with the characteristics of a large surface area and a regular pore structure, which showed excellent drug delivery capabilities for synergetic therapeutics via π–π stacking and hydrophobic interactions [5,6]. Wu et al. [7] loaded Mn into MPDA to develop a novel platform that could become a promising candidate for magnetic resonance imaging (MRI)-guided photothermal cancer therapy. Meanwhile, as a small molecule that regulates interleukin-6 (IL-6)-induced inflammatory responses, RCGD423 also was loaded into MPDA for osteoarthritis treatment by sustained and controlled drug release, according to Wang’s report [8]. Clearly, as a material with the potential for diverse medical applications, it is necessary to assess MPDA further for its biosecurity.

Previous toxicology studies focused on regular organs, such as the liver, brain, kidney, and reproductive organs; however, intestinal tissue and intestinal microflora appear to be under-investigated [9]. In recent years, researchers found that intestinal microflora comprise a complicated ecosystem, with functions that protect against pathogenic attacks, that obtain remaining energy from food, secrete critical molecules for tissue growth, and that maintain physiological inflammation of the intestine [10,11]. Thus, it is necessary to explore the impact of PTAs on the intestine and on intestinal microflora. Many toxicology studies centered on the intestinal microenvironment following the oral administration of PTAs [12,13,14,15,16]. However, some researchers found that small particles exposed to a variety of drug delivery routes could be detected in the intestine [17]; this suggested that the intestinal microbiome might also be affected by other exposure routes. Based on our previous research [18], MPDA was demonstrated to have no significant acute toxicity to most internal organs of mice who exposed to different drug delivery routes. Due to the growing recognition of the importance of intestinal microflora to health, the impact of MPDA on intestinal tissues and microflora via three kinds of drug delivery methods should be carefully explored.

Herein, we present a systematic investigation of the biocompatibility of MPDA by researching its effects on the intestines and on the intestinal microflora. Firstly, MDPA did not induce cell death in vitro, according to Cell Counting Kit-8 (CCK8) and LIVE/DEAD assays. Next, twelve female mice were randomly grouped and exposed to three different drug delivery routes, including intramuscular injection (i.m.), intravenous injection (i.v.), and intragastric administration (i.g.). According to the results of the histological analysis, there was no significant precipitation of MPDA in the intestinal tissues. Similarly, the morphology of intestinal villi and mucosa of those mice groups injected with MPDA revealed no significant changes, in comparison with the control group. Considering the importance of intestinal microflora to living organisms, we evaluated the intestinal microflora through 16S rRNA sequencing and bioinformatics analysis. According to the results of alpha and beta diversity analyses, the diversity of intestinal microflora was not affected by different administration methods. Consistently with the above results, the sample level clustering analysis and principal coordinate analysis (PCoA) also demonstrated that the diversity of each group was similar. This research supports the safety of MDPA for the intestine and intestinal microflora via different drug delivery approaches, and supports the theoretical basis for its reasonable clinical application.

## 2. Results

### 2.1. Characterization of MPDA 

MPDA was synthesized using methods that followed previous reports [19,20]. The scanning electron microscope (SEM) and transmission electron microscope (TEM) images clearly illustrate that the MPDA were homogeneous spheres with a well-ordered, mesoporous structure. The calculated average diameters of the MPDA in SEM and TEM images were 211.4 ± 22.22 nm and 195.2 ± 15.32 nm, respectively (Figure 1a,b). Then, the elemental mapping images exhibited that MPDA was mainly composed of carbon; nitrogen and oxygen also contributed to maintain the morphology of MPDA (Figure 1c). Considering that drugs were mainly loaded into MPDA through π–π stacking, the surface area was very important to MPDA’s drug loading ability. Therefore, N_2_ adsorption–desorption isotherms and the pore diameter were detected. According to Figure 1d, the Brunauer–Emmett–Teller (BET) analysis results indicated that the surface area of MPDA was 28.191 m^2^ g^−1^, which showed that the surface area of MPDA makes it a drug carrier with good loading and release capacities. Moreover, the Barret–Joyner–Halenda (BJH) data showed that MPDA had a pore diameter of 1.422 nm and a pore volume of 0.066 cm^3^ g^−1^, suggesting that the porous hollow-structured MPDA was successfully fabricated. Additionally, the X-ray powder diffraction (XRD) patterns were observed at roughly 2θ = 22° (Figure 1e), corresponding to the (002) crystal plane, and further validating that carbon was the main component of MPDA, in accordance with the element mapping results. Likewise, the wide and flat peak in XRD patterns indicated that the MPDA should be considered amorphous. Moreover, via dynamic light scattering (DLS) measurement, the hydrated particle size and particle size distribution (PDI) of MPDA were detected to be 196.25 ± 10.6 nm and 0.233 ± 0.01, respectively (Figure 1f). The diameter of MPDA detected by DLS was similar to those results calculated from SEM and TEM images. Meanwhile, the small PDI implied that there was a highly homogeneous size distribution of MPDA. Moreover, the zeta potential values of ±0–10 mV, ±10–20 mV, ±20–30 mV, and ˃±30 mV, meant that highly unstable, relatively stable, moderately stable, and highly stable, respectively, in drug delivery articles, according to Bhattacharjee’s report [21]. The zeta potential of MPDA was measured to be −30 ± 0.7 mV (Figure 1f), which indicated that MPDA was highly stable, and therefore is the reason why MPDA has broad clinical application prospects. All shown characterization data indicated that the MPDA in this study was of high quality, and approached other common MPDAs that have been widely used in biomedicine research; these findings set the stage for the following biocompatibility analysis.

### 2.2. In Vitro Cytotoxicity of MPDA 

In order to evaluate the biocompatibility of MPDA, HCT116 cells and LO2 cells were incubated by different concentrations of MPDA for 24 h. According to the results of CCK8 assays, MPDA showed no obvious cytotoxicity to HCT116 cells and LO2 cells (Figure 2a,b). Meanwhile, the confocal fluorescence images exhibited very few HCT116 cells and LO2 cells that were dead with the treatment of MPDA (Figure 2c,d). Therefore, the MPDA could be considered to be a safe biomaterial at high concentrations in vitro, providing a reliable safe range for other application studies of MPDA.

### 2.3. Histopathological Evaluation

In order to explore the effects of different MPDA delivery routes on the intestine, twelve BALB/c mice were separated into four equal groups and exposed to different drug delivery methods (Table 1). In addition, no abnormal behaviors were observed, including vocalizations, labored breathing, difficulty moving, hunching, or unusual interactions with cage mates.

The histopathological changes in the jejunum, ileum, colon, and mesenteric lymph nodes (MLN) of mice were evaluated after exposure to MDPA for 7 days. As shown in Figure 3, intestinal mucosa abscission, glandular atrophy, gland erosion, level disturbances, or fibrous tissue hyperplasia could not be observed in the overall intestinal structure of the treatment groups. In addition, MPDA did not induce significant morphological changes in the MLN in any of the different exploration routes, and no MPDA precipitated in the intestinal tissue. As a result, MPDA was not found to cause significant histological intestinal damage through different exposure routes.

### 2.4. Intestinal Microflora Distribution Analysis

Although the data of biological experiments and histological experiments demonstrated that MPDA did not cause intestinal damage, MPDA could not yet be considered to be a safe biomaterial. Based on the complex environment of intestinal microflora and the important role of intestinal microbiota in living organisms, it was necessary to elucidate the effects of MPDA on the intestinal microflora. In order to explore whether MPDA influences the intestinal microbiome in mice, we described the predominant bacterial composition landscape. Firmicutes and Bacteroidetes were the primary phyla, the percentages of which were over 70% in each group. Then, we compared the percentages of the top 7 phyla between the three treatment groups and the control group. We did not find any significant differences (*p* > 0.05) (Appendix A), which indicated that MPDA did not change the composition of bacteria at the phylum level.

We also performed the above analysis at the genus level. As shown in Figure 4b, we determined the percentages of the top 19 genera in each group. Next, a comparison of each genus was performed between the control group and the three treatment groups. No significant differences were found (*p* > 0.05) (Appendix A), which indicated that MPDA did not change the composition of bacteria at the genus level.

### 2.5. Alpha and Beta Diversity Analyses

In order to further analyze the toxic effects of MPDA among drug deliver ways, we measured the microbial diversity to investigate microbial compositions. Alpha diversity was generally evaluated by a combination of several different indexes. Here, we calculated six common alpha diversity indexes and compared the differences between the control group and other treatment groups, including Shannon, Invsimpson, richness, Chao1, Ace and Pielou. The means of the Shannon index (Figure 5a), Invsimpson index (Figure 5b), and Pielou (Figure 5c) of the control group were close to those of the i.g. and the i.m. group, and slightly higher than the i.v. group; meanwhile, there were no significant differences between the control group and other treatment groups (*p* > 0.05). For other indexes including ichness (Figure 5d), Chao1 (Figure 5e), and Ace (Figure 5f), no significant differences were found either, between control and treatment groups (*p* > 0.05). These results suggested that a certain dose of MPDA might not significantly alter the alpha diversity of the intestinal microflora among those groups.

The principal coordinate analysis (PCoA) based on the Bray–Curtis distance was a well-suited method to evaluate the intestinal microbial beta diversity between the control group and other treatment groups. The distances within each treatment group were relatively larger than those within the control group, but showed no significant differences (Figure 6a). The distribution of each point was relatively scattered as indicated in the PCoA plot (Figure 6b), and there was no obvious aggregation trend, which indicated that all of the samples from the four groups harbored similar microbial compositions. 

Furthermore, the Pearson correlation coefficients between any two samples on the basis of the microbial abundance profiles were all positive and relatively large (Figure 6c), with no obvious clustering within groups observed, indicating that the microbial profiles of all the samples with/without groups were similar to each other. Therefore, these bioinformatics results showed that the effect of MPDA on the intestinal microbiome was minimal, among the different means of administering it.

### 2.6. Differential OTUs Analysis

In order to see whether any microbiome was over- or under-represented in the intestine, we performed a differential abundance analysis between the control group and each treatment group (Appendix A). No significantly different operational taxonomic units (OTUs) between the control group and the other treatment groups were observed (*p*.adj > 0.05). Taken together, these results support the view that MPDA does not alter the intestinal microflora significantly in mice after 7 days of treatment.

## 3. Discussion

The intestine is the largest interface in the body that has direct contact with the external environment [11]. Abundant bacteria settle in the intestinal tract of humans, establishing the mucosal immune system, and offering microbial antigens and metabolites [22]. Thus far, growing evidence suggests that there is a complex relationship between bodily health and the intestinal microflora. For example, inflammatory bowel disease (IBD) results from an abnormal innate mucosal immune response to the intestinal microflora in a susceptible host [23]. According to the results of the metagenomic approach, C Manichanh et al. [24] found that those people with Crohn’s disease had a lower intestinal microbial diversity than that of healthy people, especially among the Firmicutes. Indeed, intestinal microflora offer a new perspective for understanding causation and pathogenesis, and also play a significant role in human health. 

Photothermal therapy with the assistance of efficient PTAs is considered to be an indispensable strategy in the biomedical field [25,26]. Therefore, it is important to evaluate the biocompatibility of PTAs. As a PTA in clinical applications, MPDA has been recognized as a promising carrier for drug delivery due to its high surface area ratio, mesoporous structure, and easy functional modification; it has also been widely used in disease research. Huang et al. [27] fabricated a biomimetic nano-platform by MPDA loading the autophagy inhibitor chloroquine (CQ), which blankets homologous prostate cancer cell membranes (CMs) for homotypic targeting effects. Compared with other groups, the MPDA@CMs-CQ group showed remarkable tumor ablation ability after irradiation. Moreover, MPDA also contributed to neuropathic pain, according to Kuthati et al. [28]. They creatively made a novel nanomaterial that combined with MPDA and morphine to resisting the production of reactive oxygen species (ROS), and suppressed antioxidant genes, resulting in morphine antinociceptive tolerance (MAT). Given the promising clinical application of MPDA, in conjunction the importance of intestinal microflora to the human body, it was critical to evaluate the intestinal toxicity of MPDA and to explore its possible influences on intestinal microflora.

In this research, we systematically investigated the toxicity of MPDA on the intestine and on the intestinal microflora. Firstly, to ensure that the MPDA used in this research was of high quality and common, MPDA was detected by a variety of characterization tests; its positive results laid a foundation for subsequent experimentation. Secondly, we explored toxicity of MPDA at different concentrations in HCT116 cells and LO2 cells. The results revealed that cell viability did not change at 100 μg/mL. The same conclusion also was observed in fluorescence images, in which MPDA could not induce cell death, even at 200 μg/mL and 400 μg/mL, which demonstrated a reliable safe range for other applicational studies for MPDA. Furthermore, MPDA did not induce pathologic damage to the intestine, according to HE staining images, and the histological structure of MLN also showed that an acute immune response could not be activated by MPDA for different drug delivery methods. In order to further explore the effect of MPDA on intestinal microflora, we investigated changes in the intestinal microbial community based on predominant analysis, alpha and beta diversity analyses, cluster analysis, as well as differential abundance analysis, using 16S rRNA gene sequencing. Comparing the different methods of administering MPDA, we found that the intestinal bacterial composition shared similar microbial signatures. In support of this, no significant differential bacteria were found when comparing all treatment groups to the control group at either a phylum or genus level. In short, MPDA in large doses did not show obvious toxicity to the intestinal tracts and microflora in mice using three different drug delivery methods, which sets the stage for treatments using MPDA in intestinal diseases.

## 4. Materials and Methods

### 4.1. Materials

1,3,5-trimethylbenzene (TMB, AR, 97%), dopamine hydrochloride (98–100%), and ethanol (AR, 95.0%) were purchased from Aladdin Reagent (Shanghai, China). Ammonia aqueous solution (NH_3_·H_2_O, AR, 25–28%) was acquired from Macklin (Shanghai, China). Pluronic F127 (contains 100 ppm BHT) was purchased from Sigma-Aldrich (St. Louis, MO, USA). Human colon cancer cell line (HCT116 cells) and normal liver cell (LO2 cells) were purchased from ATCC (Richmond, VA, USA). The CCK8 kit was purchased from Dojindo (Kumamoto, Japanese). A LIVE/DEAD staining kit was purchased from Invitrogen (Carlsbad, CA, USA). Penicillin–streptomycin, Dulbecco’s Modified Eagle Medium (DMEM), fetal bovine serum (FBS), and phosphate buffered saline (PBS) were acquired from Gibco (Shanghai, China). A hematoxylin and eosin (HE) staining kit was purchased from Solarbio (Beijing, China). 

### 4.2. Synthesis of MPDA 

Dopamine hydrochloride (0.3 g) and F-127 (0.2 g) were dissolved in an ethanol and water mixed solvent (1:1 by vol), and TMB (320 μL) was added slowly. Ultrasonic stirring was applied until the mixed solvent turned milky white. Next, 750 μL of NH_3_·H_2_O was added, and the mixed liquid began to turn brown. Then, the solutions were stirred (1000 rpm) for 2 h at room temperature, and the MPDA was accumulated by centrifugation (12,000 rpm, 10 min). At last, the dark product was washed using ethanol and pure water respectively, then dried at 50 °C overnight to obtain MPDA power.

### 4.3. Characterization

The morphology and elemental distribution of MPDA were measured using a transmission electron microscope (TEM, JEOL JEM-2100F, Tokyo, Japanese) at 200 kV. The morphology of MPDA was detected by a scanning electron microscope (SEM, FEI Apreo HiVac, Waltham, MA, USA) at 5 kV. 

Nitrogen adsorption and desorption isotherms were measured using an automated gas sorption analyzer (Autosorb-iQ, Miami, FL, USA) at a temperature of 77.35 K, and the specific surface area was calculated using the BET (Brunauer–Emmett–Teller) method when the nitrogen relative pressure was in the range of 0.05–0.35. The pore size distribution and pore volume were calculated from the desorption branch of the isotherm, according to the Barrett–Joyner–Halenda (BJH) model.

X-ray diffraction (XRD) patterns of samples were recorded via diffractometer (Rigaku D/max 2200 pc, Tokyo, Japanese), with Cu Kα radiation of wavelength λ = 0.15418 nm, with Cu Kα radiation of wavelength λ= 0.15418 nm, at 40 kV and 40 mA. The scanning rate of the XRD patterns was 0.1 °/s, and the range of 2θ was from 0° to 80°.

MPDA was diluted to a 10 μg/mL solution with ultrapure water (pH = 7) for the particle size and zeta potential measurement via dynamic light scattering (DLS, Malvern Zetasizer Nano ZS, Malvern, UK).

### 4.4. Cell Culture

LO2 cells and HCT116 cells were cultured in DMEM medium with 10% FBS and 1% PS. All cells were cultivated at 37 °C under a moist environment with 5% CO_2_.

### 4.5. Cell Viability Assay 

LO2 cells and HCT116 cells were seeded and cultured in DMEM medium in 96-well plates. Next, the old culture medium was supplanted by fresh culture medium with different concentrations of MPDA. After incubating for 24 h, Cell Counting Kit-8 (CCK8) solution in DMEM medium was added slowly, following corresponding protocols, and the plates were incubated for another hour. At last, the absorbance value was detected using a microplate reader (Varioskan Lux, Thermo Scientific, Waltham, MA, USA) at 450 nm.

### 4.6. Live/Dead Cell Staining Experiments

The cells were seeded and cultured for 24 h for cell attachment. Then, the cells were incubated with MPDA for 24 h. Thereafter, Calcein AM and pyridine iodide (PI) were used to stain the cells following the protocols. A confocal laser scanning microscope system (LSM880, Zeiss, Oberkochen, Germany) was used to detect fluorescence.

### 4.7. Animal Experiments

The Guangdong Medical Laboratory Animal Center (Guangdong, China) provided twelve BALB/c female mice, which were maintained in clear rooms for 7 days in order to acclimate them to a new environment with 12-h light/dark cycles. The temperature was maintained at 20 ± 3 °C in the animal room. The animals were accommodated 3/cage, and they were provided a standard chow pellet diet and clear drinking water to ensure their health. The experiment was approved by animal ethics guidelines of the Animal Ethical and Welfare Committee of The Eighth Affiliated Hospital, Sun Yat-sen University. 

Before the experiments were initiated, all BALB/c mice were separated into four groups, randomly and equally (Table 1). The mice that belonged to treatment groups were exposed to MPDA once. The behavior of every mouse was observed every day for 7 days. Next, all of the mice were euthanatized via cervical dislocation, and their faeces, intestinal tracts, and mesenteric lymph nodes (MLN) were collected for histopathological examinations and 16S rRNA gene sequencing.

### 4.8. Histopathological Examinations

After the intestines were thoroughly rinsed, the fresh tissues were immediately fixed in 10% formalin solution. After 24 h, those fixed tissues were enclosed by paraffin, then sectioned along the longitudinal axis. All of the tissue sections were stained carefully using hematoxylin and eosin (H&E) staining kits, and images were taken using standard light microscopy (Axio Observer, Zeiss, Oberkochen, Germany), and all images were collected at a magnification of 250X.

### 4.9. DNA Extraction and Sequencing of 16S rRNA Gene Amplicons

Total genomic DNA from 12 stool specimens from the 12 BALB/c mice was extracted on the basis of the CTAB/SDS method. Then, the V3–V4 region of the 16S rRNA gene was PCR-amplified using forward and reverse fusion primers, with the forward primer consisting of an 8-base pair (bp) bar code, and the 341F primer (CCTAYGGGRBGCASCAG). The reverse fusion primer included an 8-base pair bar code and the 806R primer (GGACTACNNGGGTATCTAAT). The sequencing was performed at an Illumina NovaSeq platform, and 250 bp of paired-end reads were created. The quality of sequencing data is presented in Appendix A.

### 4.10. Bioinformatics Analysis of 16S rRNA Sequencing

First, raw data was split through blocking the barcode as well as the primer sequence; paired-end reads were merged to assemble sequences using FLASH (v1.2.7). Then, according to the QIIME (v1.9.1), high-quality sequences were obtained by quality filtering. After removing the chimera sequences against the Silva database (https://www.arb-silva.de/, accessed on 25 October 2021) using the UCHIME algorithm, clean data were finally obtained.

In order to classify microorganisms, we merged all clean sequencing data from 12 samples into one file, filtered the poor-quality sequences with parameter maxee < 1, and searched the unique sequences using USEARCH [29]. After reads were prepared, unique sequence clustering was performed using a greedy clustering method with USEARCH. A feature table containing operational taxonomic units (OTUs) and associated abundance was generated using merged reads with 97% similarity in VSEARCH (v2.14.2) [30]. In order to further match microbial information to the OTUs, the taxonomies were predicated against the Ribosomal Database Project database [31] using VSEARCH.

For each taxonomic level, the OTUs table of absolute abundance was normalized to one of relative abundance. Alpha and Beta diversity, as well as Pearson correlation coefficients, were calculated with relative abundance of bacterial genus. Alpha diversity indices, including Shannon index, Invsimpson index, richness, Chao1, Pielou, and ACE, were calculated using the vegan package in R. The *p*-value of Alpha diversity comparison between control and treatment groups was analyzed using the Wilcoxon rank sum test. Principal coordinate analysis (PCoA) based on the Bray–Curtis distance were assessed to compare the microbial community differences using vegan and ape package in R. The differential microbiota between the control and treatment groups were calculated with relative abundance using the Kolmogorov–Smirnov test in R, and the *p*.adjust function was used to adjust *p*-values for multiple hypothesis calculated with Benjamini–Hochberg method.

### 4.11. Statistical Analysis

All results were expressed as means ± standard deviations (SD), and statistical differences between any two groups were examined using *t* test. A difference was considered statistically significant for *p*-value < 0.05.

## 5. Conclusions

In summary, all results indicated that MPDA could not induce obvious toxicity, in vitro or in vivo. Although MPDA was administered in three different ways, intestinal morphology and intestinal microflora did not significantly change in comparison with the control. These findings provide new evidence for the safety of MPDA for clinical applications.

## Figures and Tables

**Figure 1 molecules-27-06461-f001:**
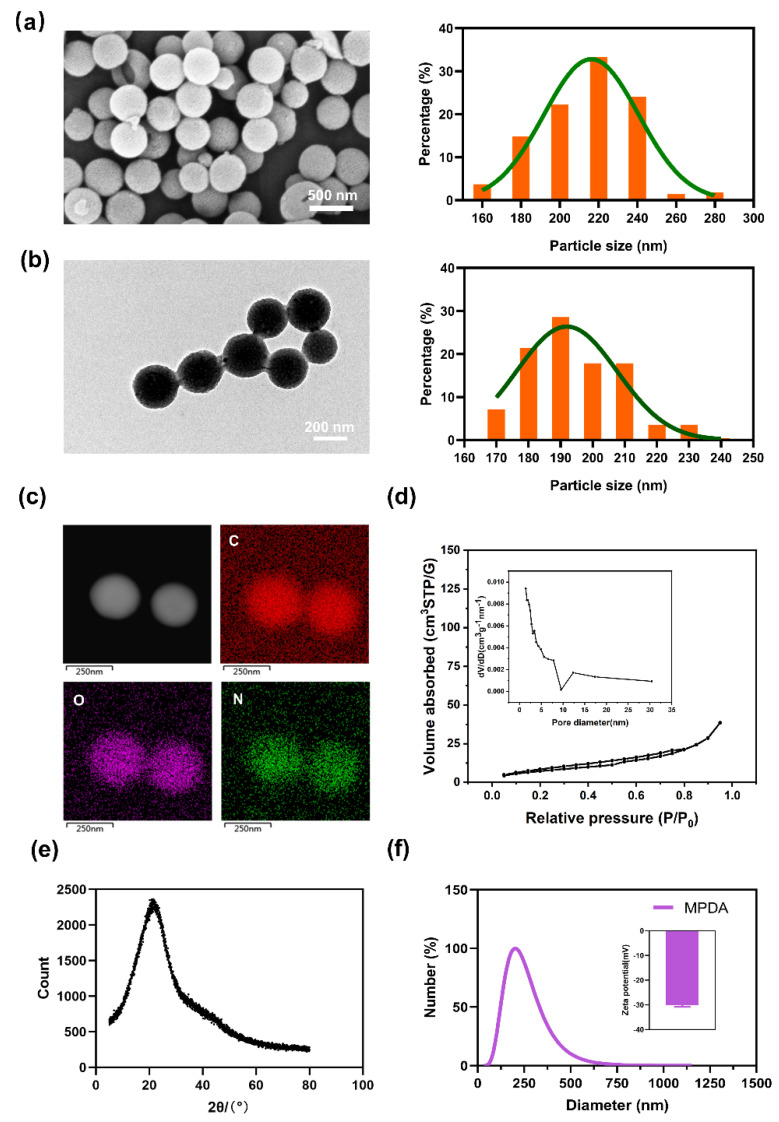
Characterization of MPDA. (**a**) SEM image and the SEM-extracted diameter distribution histogram. (**b**) TEM image and the TEM-extracted diameter distribution histogram. (**c**) Elemental mapping of MPDA. (**d**) N_2_ adsorption–desorption isotherms and the pore diameters of MPDA. (**e**) XRD patterns. (**f**) Size distribution and zeta potentials.

**Figure 2 molecules-27-06461-f002:**
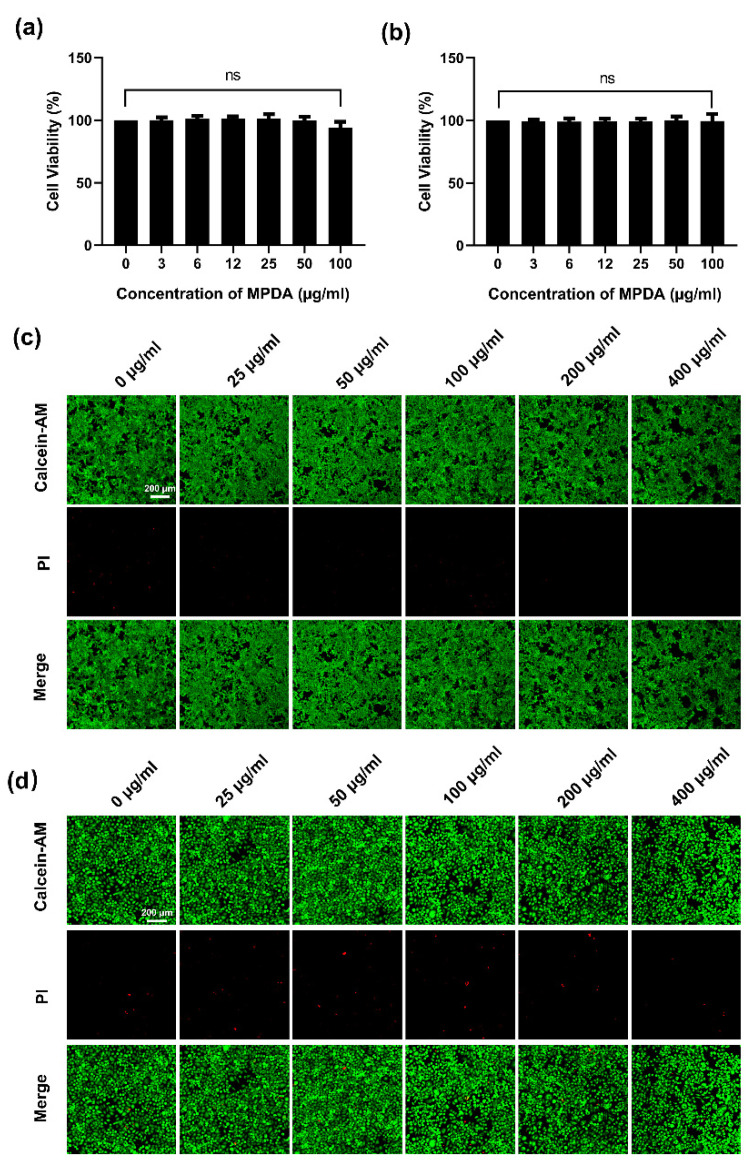
Effect of MPDA on cell viability. Comparison of different concentrations of MPDA in (**a**) HCT116 cells and (**b**) LO2 cells. Fluorescence images of (**c**) HCT116 and (**d**) LO2 after treating with different concentrations of MPDA for 24 h. Scale bar: 200 μm.

**Figure 3 molecules-27-06461-f003:**
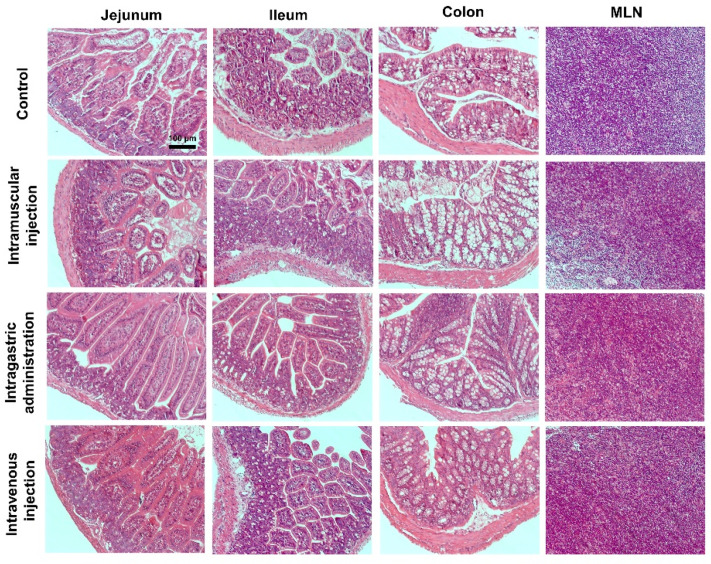
Histological examination of the jejunum, ileum, colon, and mesenteric lymph nodes (MLN) from the BALB/c mice exposed to intramuscular injection, oral administration, and intravenous injection at 7 days. Scale bar: 100 μm.

**Figure 4 molecules-27-06461-f004:**
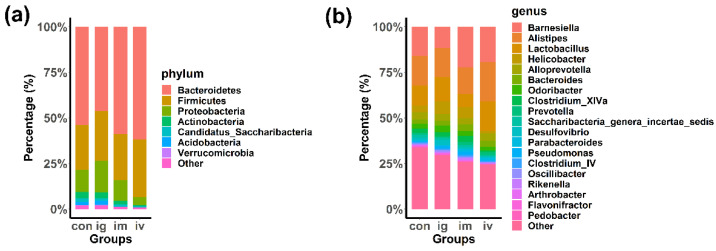
Predominant bacteria identified in the intestinal tissues of mice. (**a**) Intestinal microflora dispersion at the phylum level; (**b**) intestinal microflora dispersion at the genus level.

**Figure 5 molecules-27-06461-f005:**
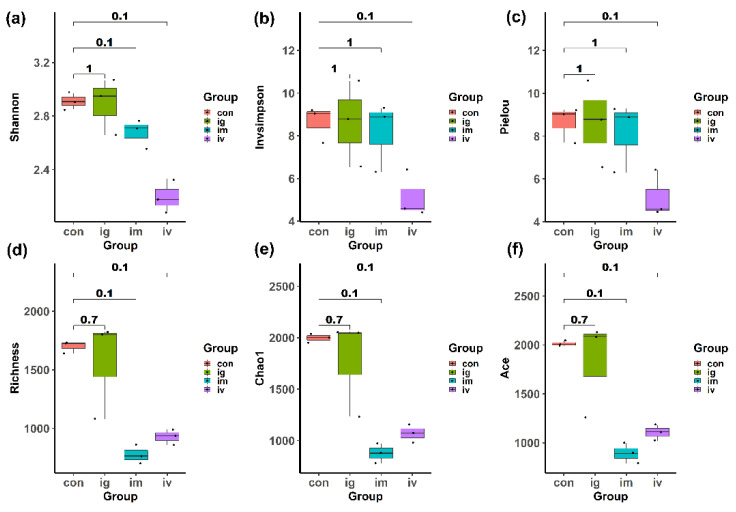
MPDA caused no significant differences in the intestinal bacterial community of Alpha diversity. (**a**) Shannon index; (**b**) Invsimpson index; (**c**) Pielou index; (**d**) richness; (**e**) Chao1 index; (**f**) Ace index. Wilcoxon rank sum test was used to compare control and treatment groups, and the *p*-value threshold was set at 0.05 for statistical significance.

**Figure 6 molecules-27-06461-f006:**
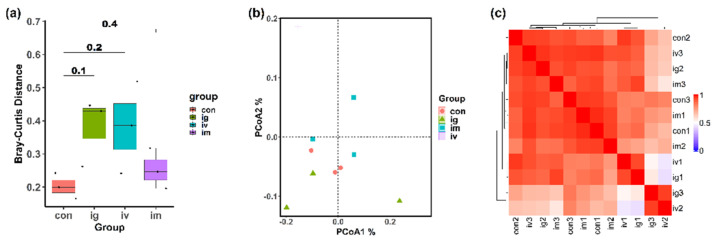
MPDA caused no significant differences in intestinal bacterial communities among control and treatment groups. (**a**) Bray−Curtis distance; (**b**) PCoA analysis; (**c**) heat map showing the clustering of samples based on Pearson correlation coefficient of genus. Wilcoxon rank sum test was used to compare control and treatment groups, and the *p*-value threshold was set at 0.05 for statistical significance.

**Table 1 molecules-27-06461-t001:** Mice groups exposed by three different drug delivery routes.

Group	CTL	i.m.	i.g.	i.v.
**Dose**	0 mg	40 mg	50 mg	8 mg
**Mice number**	3	3	3	3
**Period**	7 Days	7 Days	7 Days	7 Days
**Abnormal behavior ***	No	No	No	No

* Abnormal behaviors including vocalizations, labored breathing, difficulty moving, hunching, or unusual interactions with cage mates.

## Data Availability

The 16S rRNA gene sequencing from the current study is available at the National Genomics Data Center, with the accession number PRJCA010755 (https://ngdc.cncb.ac.cn/bioproject/browse/PRJCA010755, accessed on 15 July 2022). All data generated and analyzed in this study are included in this published article and its Appendix A.

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
