# Peer review of "Toxicity Analysis of Mesoporous Polydopamine on Intestinal Tissue and Microflora"

_molecules, 2022, doi:10.3390/molecules27196461_

Round 1
Reviewer 1 Report
General issues:
The authors report on the synthesis and characterization of MPDA NPs that have been recognized as a promising carrier for drug delivery purposes due to their physicochemical properties. In this work a preliminary study on toxicity of MPDA NPs on intestinal tissues and gut flora has been carried out. No results are presented on drug release. I think the authors should improve their work by introducing an evaluation of the properties of their particles on release study.
Specific issues:
As well pointed out very well by the authors in the introduction section, nanomaterials have unique physico-chemical properties and can be largely applied in many biotechnological fields. However, one of the most important concerns regards their safety evaluation on the basis of their ability to be applied for clinic purposes. The authors report on Mesoporous Polydopamine Nanoparticles (MPDA NPs) as polymer based materials with large surface area and regular pore structure able to enhance their drug delivery properties in particular in medical field. On this basis the authors have primarily focused their activity on the evaluation of MPDA NPs safety when applied on intestinal tissues and gut microbiota they consider up to now underappreciated targets.
An important issue is the novelty and the impact of the current work. I think that the authors should provide experiments demonstrating the use of mesoporous polydopamine nanoparticles as drug delivery vehicles, already known and studied by other authors, on intestinal flora and tissues.
They reported on the physico-chemical characterization of MPDA NPs but they should provide more information on the conditions they performed the experiments adding more details regarding the parameters used to carry out the activity in order to make possible their reproducibility (see NPs characterization section in the manuscript)
Finally the dimensions of the MDPAs range from 250 to 500nm and therefore cannot be considered nanometric. Generally, nanometric materials have dimensions less than 100 nm. Please correct in the text. A more detailed description of the SEM micrographs should be added.
Please revise the written English and correct typo errors overall the manuscript
Reviewer 2 Report
The current work focuses on the Toxicity analysis of mesoporous polydopamine nanoparticles on intestinal tissue and microflora. The author’s some effort into the manuscript, but major issues should be addressed. The novelty and the impact of the current work are low, where mesoporous polydopamine nanoparticles were already prepared and investigated their toxicity previously.
https://jnanobiotechnology.biomedcentral.com/articles/10.1186/s12951-021-01131-9
https://aip.scitation.org/doi/full/10.1063/5.0088447
In addition, the main problem in the manuscript is that the authors show only results without interpretations of it or confirmation by citation. More details are required to explain the obtained results.
Abstract
- The abstract is very general information (Line 10-17), please rephrase it with the novelty and focus on the main outcomes (Line 18-20) that the manuscript will deal with it.
Introduction
- The introduction provides sufficient background, and all relevant references are included. But, the novelty of this work is not highlighted and it was not clear the author's contribution in comparison to other previous works.
Materials and Methods
- The used materials with their impurities should be inserted
- Line 271, correct typo NH3•H2O
- Experimental part required rephrasing to be more precise with details and logic to the reader for reproducibility. No information about the used amount of ethanol and water mixed solvent, what is the temperature of the reaction, stirring type and speed of it,…..
- Line 286-292, Characterization of MNP
“The morphology and elemental distribution of MPDA NPs were measured by the
transmission electron microscope (TEM). The morphology of MPDA NPs was detected by
the scanning electron microscope (SEM). The particle size and zeta potential of MPDA
NPs were measured by dynamic light scattering (DLS). Surface area and pore diameter
were detected by brunauer-emmett-teller (BET) measurements and the crystal structure
was analyzed by the X-ray powder diffraction (XRD).”
What is the model version of each instrument? What is the country manufacture?
What is the condition of measurements?
What is the condition analysis for XRD, and scanning rate? 2θ range?
What is the instrument version used to measure DLS and zeta-potential!!
Result and discussion
- One of the main problems in the manuscript is that the authors show only results without interpretations of it or confirmation by citation. More details are required to explain the obtained results.
-Line 87-89, It state “The scanning electron microscope (SEM) and transmission electron microscope (TEM) images clearly illustrated that the MPDA NPs was homogeneous sphere with well-order mesoporous structure”
No details or information mentioned on size and particle size distribution obtained from SEM TEM?
-Line 84-95, It state “brunauer-emmett-teller (BET) analysis results indicated that the surface area of MPDA NPs was 28.191 m2 g−1, which provides an ideal superficial area for the drug delivery.”
Not mentioned why this result is ideal. No citation is provided to support this claim.
-Could we investigate and calculate the size from XRD and compare it with the size obtained from TEM, SEM, and DLS?
-Zeta potential show -30±0.7mV !!! what information I can gain from this number? Is good for nanoparticle stability for biomedical applications or not? Why is the sign negative?
-What is the condition for DLS analysis, and solvent? Concentration? pH?
Conclusion
Where conclusion section!!!!!
References
- Correct the reference with complete citation e.g. no.12.
- make one system style and insert the more recent related citation.
Round 2
Reviewer 1 Report
The authors answered all the questions arised and they provide a final version of the submitted paper substantially improved.
Reviewer 2 Report
All issues are solved. Accept in the present form